# Development and validation of a self-administered computerized cognitive assessment based on automatic speech recognition

**Hyun-Ho Kong[1,2], Kwangsoo Shin[3], Dong-Seok Yang[4], Aryun Kim[5,6], Hyeon-Seong Joo[7], Min Woo Oh[2]☯\*, Jeonghwan Lee[8]☯\***

1 Department of Rehabilitation Medicine, Chungbuk National University Hospital, Cheongju, Republic of Korea, 2 Department of Rehabilitation Medicine, Chungbuk National University College of Medicine, Cheongju, Republic of Korea, 3 Graduate School of Public Health and Healthcare Management, Songeui Medical Campus, The Catholic University of Korea, Seoul, Republic of Korea, 4 Technology Strategy Center, Neofect, Seongnam, Republic of Korea, 5 Department of Neurology, Chungbuk National University Hospital, Cheongju, Republic of Korea, 6 Department of Neurology, Chungbuk National University College of Medicine, Cheongju, Republic of Korea, 7 Department of Physical Therapy, Daejeon University, Daejeon, Republic of Korea, 8 Department of Psychiatry, Chungbuk National University Hospital, Cheongju, Republic of Korea

☯ These authors contributed equally to this work.
\* omo11017@naver.com (MWO); jeonghwan@cbnuh.or.kr (JL)

**Data Availability Statement:** The data underlying this study cannot be shared publicly due to the

## Abstract

Existing computerized cognitive tests (CCTs) lack speech recognition, which limits their assessment of language function. Therefore, we developed CogMo, a self-administered CCT that uses automatic speech recognition (ASR) to assess multi-domain cognitive functions, including language. This study investigated the validity and reliability of CogMo in discriminating cognitive impairments. CogMo automatically provides CCT results; however, manual scoring using recorded audio was performed to verify its ASR accuracy. The mini–mental state examination (MMSE) was used to assess cognitive functions. Pearson's correlation was used to analyze the relationship between the MMSE and CogMo results, intraclass correlation coefficient (ICC) was used to evaluate the test-retest reliability of CogMo, and receiver operating characteristic (ROC) analysis validated its diagnostic accuracy for cognitive impairments. Data of 100 participants (70 with normal cognition, 30 with cognitive impairment), mean age 74.6±7.4 years, were analyzed. The CogMo scores indicated significant differences in cognitive levels for all test items, including manual and automatic scoring for the speech recognition test, and a very high correlation (r = 0.98) between the manual and automatic CogMo scores. Additionally, the total CogMo and MMSE scores exhibited a strong correlation (r = 0.89). Moreover, CogMo exhibited high test-retest reliability (ICC = 0.94) and ROC analysis yielded an area under the curve of 0.89 (sensitivity = 90.0%, specificity = 82.9%) at a cutoff value of 68.8 points. The CogMo demonstrated adequate validity and reliability for discriminating multi-domain cognitive impairment, including language function, in community-dwelling older adults.

presence of potentially sensitive information. However, data are available for researchers who meet the criteria for access to confidential data. Interested researchers may contact the Chungbuk National University Hospital Ethics Committee for data access inquiries at TEL: +82-43-269-6771.

**Funding:** This research was supported by the Medical Device Technology Development Program (grant number: 20014701, modular quantitative aging assessment and care service based on multiple sensors for aging in-home) funded by the Ministry of Trade, Industry, and Energy (MOTIE, Sejong, Republic of Korea).

**Competing interests:** The authors have declared that no competing interests exist.

## Introduction

Life expectancy is constantly increasing owing to advances in medical technology and improvements in overall healthcare [1]. However, with an aging population, the number of older adults with cognitive impairment (CI) is increasing. Dementia affected an approximately 57.4 million people worldwide in 2019, and this figure is expected to rise to 83.2 million by 2030 and 152.8 million by 2050 [2]. Older adults with CI have difficulties in independently performing daily activities [3], experience social isolation and withdrawal [4], and suffer from a decrease in their overall quality of life [5]. Additionally, families and society are burdened by the medical and daily care required for older adults with CI. The World Health Organization has estimated that the global cost of dementia was $1.3 trillion in 2019, which is expected to increase to $2.8 trillion by 2030 [6].

The annual incidence rate of individuals with normal cognition transitioning to Alzheimer's disease (AD) or mild cognitive impairment (MCI) is 1–2% [7], whereas the incidence rate of dementia among individuals with MCI is 8–15% [8]. Furthermore, a study showed that approximately 80% of all patients with MCIs are eventually diagnosed with dementia based on their six-year follow-up data [9]. Because there is no commercially available disease-modifying treatment for dementia and only symptomatic treatments are available [10], reducing modifiable risk factors and appropriate lifestyle counseling through early detection of MCI are the only known methods for slowing the progression of MCI to dementia [11].

Clinicians diagnose CI through comprehensive evaluations by both observing the clinical features of patients with CI and through neuropsychological, laboratory, and neuroimaging tests [12]. However, the various neuropsychological tests used to diagnose CI have limited use in general clinical settings because they must be administered by trained professionals to ensure accurate assessments [13]. Therefore, simple cognitive screening assessment tools such as the mini–mental state examination (MMSE) and Montreal cognitive assessment (MoCA) are widely used to screen for CI [14]. In particular, the MMSE has been translated and adapted into various languages across different countries, with multiple validation studies demonstrating its high diagnostic value for detecting CI, regardless of language differences [15].

With advancements in digital healthcare, attempts have been made to evaluate cognitive function using computerized cognitive tests (CCTs) [16]. These tests have several advantages including lower examiner bias, the ability to be self-administered or by testing technicians rather than trained professionals, and the convenience of automatic scoring and storage of the results [16, 17]. However, most CCTs developed thus far use touch screens or mouse/keyboard input interfaces, which are difficult to use by older adults who are uncomfortable using digital devices. Additionally, it is difficult to assess higher cognitive functions, such as language comprehension/expression and language-related executive processes, because these tests do not include speech recognition. Furthermore, many CCT methods use only visual instructions, which have limitations in that they are challenging for illiterate individuals [16, 17].

Previous studies have demonstrated the potential of automatic speech recognition (ASR) for detecting CI [18–20]. ASR is a technology that converts spoken language into text using advanced algorithms, including natural language processing and machine learning [21]. Some studies have utilized ASR to extract acoustic and linguistic features from spontaneous speech, achieving promising accuracy in distinguishing CI from healthy controls and even mild AD in certain cases [18, 19]. Other research has focused on using ASR to automate the scoring of language fluency tasks in assessments like the MoCA, highlighting its applicability in non-English-speaking populations [20]. The integration of ASR in cognitive assessments primarily facilitates the evaluation of language-related cognitive functions, as well as enhances objectivity, enables real-time analysis, and improves accessibility for individuals with limited literacy

[18, 22, 23]. However, previous studies have primarily focused on isolated linguistic or acoustic features, lacked comprehensive multi-domain cognitive assessments, and frequently depended on complex preprocessing techniques, which may limit their scalability and broader applicability in various clinical settings [18–20].

Therefore, we developed CogMo, a CCT designed to overcome the limitations of traditional cognitive assessments. CogMo evaluates language-related cognitive functions using a voice interface with ASR, ensuring accessibility for older adults, including those with limited literacy. It also organizes test content into familiar topics to enhance engagement, making the test more user-friendly for older populations. This study aimed to determine the validity and test-retest reliability of the CogMo for assessing cognitive functions compared with the MMSE traditionally used in clinical settings.

## Materials and methods

### Study population

This study recruited community-dwelling older adults aged ≥65 years who visited a local dementia care center for cognitive function assessment between March and October, 2023. The inclusion criteria were as follows: ≥65 years and understood the study content and voluntarily agreed to participate. The exclusion criteria were as follows: i) individuals with neurological or musculoskeletal conditions that make it difficult to operate a tablet personal computer (PC) using their fingers (e.g., hemiplegia due to stroke, upper extremity fractures, or amputations), ii) those with visual impairments assessed using Dr. Hahn's standard vision chart based on the Snellen chart [24], and iii) those with hearing impairments identified through a verbal instruction test in which participants were asked to repeat simple words or sentences spoken at a conversational volume in a quiet environment [25]. Individuals with blindness or those who were unable to follow the verbal instruction test due to hearing impairments were excluded, as these conditions could interfere with the accuracy of cognitive assessments using a CCT. Finally, 102 participants were recruited, excluding those who had difficulty performing the CCT owing to hearing impairment (n = 1) and hemiplegia (n = 1), 100 participants (normal cognitive function group = 70 vs. CI group = 30) were included in the analysis. The data for this study were collected directly by the research team in compliance with ethical guidelines approved by the Institutional Review Board (IRB) of a tertiary hospital (IRB number: 2022-08-004). Written informed consent was obtained from all participants before data collection, ensuring their anonymity and confidentiality. All procedures for data collection and analysis adhered to relevant ethical and legal standards.

### Evaluation and definition of cognitive impairment

The K-MMSE-2 (Korean, second edition, standard version) was administered as a neuropsychological test to evaluate the participants' cognitive functions. The K-MMSE-2 assesses several cognitive characteristics, such as orientation, memory, attention, and language, to screen for dementia or evaluate the severity of CI, with a total score ranging from 0 to 30 and a standardized score corrected according to the age and education level of the participant [26]. In this study, individuals with CI were defined as those whose K-MMSE-2 scores corrected for age and education level were below the mean—1.0 standard deviation (SD) compared with healthy individuals [8, 27].

### Development of computerized cognitive test

The CogMo test items were developed by a panel of clinical experts (psychiatrists, neurologists, and physiatrists) and is an Android-based application developed by computer engineers for on

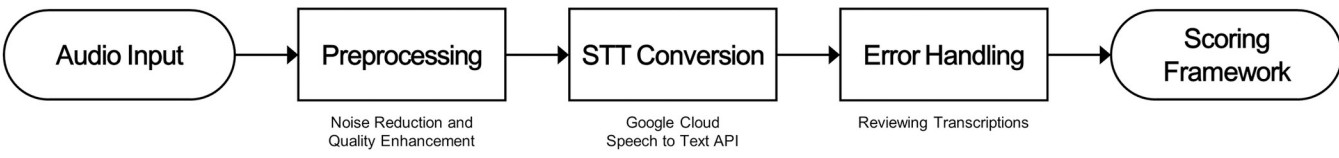

**Fig 1. Flowchart of the autonomic speech recognition processing.**

an Android-based tablet PC. CogMo utilizes the Google Cloud's speech-to-text application programming interface (API), a widely used speech recognition tool powered by advanced deep neural networks, to transcribe spoken responses into Korean text with high accuracy. This API is specifically optimized to support Korean, ensuring precise recognition of linguistic nuances and phonetic variations inherent to the language [28]. The process for ASR-based cognitive assessment includes audio input, noise reduction preprocessing, speech-to-text conversion, error validation to ensure transcription accuracy, and scoring through the CogMo framework (Fig 1). The voice data of the participants were collected during the test using the built-in microphone of the tablet PC, ensuring high-quality audio input for reliable processing. The participants performed the CCT by touching the screen of tablet PC or verbally answering the test questions, and their responses were scored automatically. Verbal responses were transcribed using the Google Cloud's speech-to-text API and automatically scored through the CogMo scoring framework, which evaluated the accuracy and relevance of the responses based on predefined criteria.

The CogMo assesses various cognitive domains, including attention, visual perception, memory (verbal and visual), execution, and language. It consists of eight items (ten subtests in total), each designed to evaluate specific aspects of cognitive function. To calculate the total CogMo score (0–100), the raw scores for each subtest were converted to a standardized scale ranging from 0 to 10 points, reflecting the relative importance and cognitive demands of each subtest. The scoring framework was based on validated tools such as the MMSE and MoCA, and the content validity index methodology was utilized [29], with seven clinical experts evaluating the representativeness and relevance of the subtests. A detailed description of the tests is presented in Table 1 and Fig 2.

### Testing procedures for the computerized cognitive test

The CogMo test was conducted in a quiet, properly illuminated room with minimal distractions. During the test, participants sat comfortably in front of a standardized tablet PC with a built-in microphone, following on-screen prompts and responding accordingly. The CogMo test is designed to be self-administered, with all instructions provided both visually and verbally. These instructions were repeated as necessary until participants fully understood the tasks. The examiner was present but intervened only if assistance was explicitly requested. Tablet settings, including volume and microphone sensitivity, were standardized across sessions to ensure consistent and reliable data collection. The total test time was approximately 15–20 min. To measure the accuracy of speech recognition and automated scoring, voice-response tests were manually scored by two independent, blinded raters who listened to the audio recorded during the CCT.

### Statistical analysis

Depending on the type of data, continuous variables are presented as means with SD, and categorical variables are presented as numbers with percentages. Before conducting the main

**Table 1. Description of the content of each test item in CogMo.**

| Item | Content |
|---|---|
| 1) Finding a puppy (Attention) | The puppy you were raising ran away from home, and you must find your dog from among various animals. First, registration was performed by showing a picture of the puppy to the participant. Next, five animals—puppy, rat, chicken, rabbit, and frog—were shown on the screen for 1.2 s in a random order, and the participants were instructed to touch the screen when they saw the puppy. After two practice trials, the test was administered for 100 s, and 1 point was awarded per correct answer. |
| 2) Matching the shadows (Visuospatial perception) | First, figures of different colors and shapes were shown, which were then darkened to shadows and superimposed. The participant was instructed to find the overlapping figures. The participants were allowed to first practice up to three times. They were awarded two points per correct answer. |
| 3) Ordering the lights (Visual memory) | A total of nine light bulbs are arranged in a square shape in three rows horizontally and three rows vertically. The bulbs were turned on and off in a random order, and the participant was instructed to remember the order in which the bulbs were turned on and touch them in that order or vice versa. If the participant failed at any step, the test for that step was repeated, and if they failed a second time, the test was terminated. |
| Forward | Touching the bulbs in the order they were turned on. Depending on the level of difficulty, from two to seven light bulbs turn on and off sequentially. The participants were awarded two points per correct answer. |
| Backward | Touching the bulbs in the reverse order in which they were turned on. Depending on the level of difficulty, from two to seven light bulbs turn on and off sequentially. The participants were awarded two points per correct answer. |
| 4) Counting numbers (Execution) | The Stroop test was administered using dice to allow testing participants who were illiterate. The number of dice ranged from one to three, and the number of dots on each dice also ranged from one to three. The numbers of dice and dots were randomly arranged. The participants were allowed to take one practice test. |
| Dots | Count the number of dots on the dice 50 times for 2 min. They were awarded 0.3 points per correct answer. |
| Dices | Change the rules, count the number of dice, and do this 50 times for 2 min. They were awarded 0.3 points per correct answer. |
| 5) Speaking sentence (Auditory memory) | Before performing the "Finding a puppy" test, registration was attempted by instructing the participant repeat the following sentence up to three times. "(Youngsoo) went to the (park), ate (kimbap), and played (soccer) since (2:00)." After completing the "Counting numbers" test, try to recall and recognize the registered sentences. |
| Recall | The participants were instructed to repeat the registered sentence from memory. The correct answer (five words) was automatically checked by ASR, and the participant was awarded three points per correct answer. |
| Recognition | Recognition was tested through five multiple-choice tests (three choices per test), three points were awarded per correct answer. |
| 6) Finding hidden money (Visual memory) | After registering for the "Speaking sentence" test, this test is registered. The screen showed a room with five household items: bookshelf, bed, picture frame, refrigerator, and desk, and on the right side of the room, four types of banknotes used in Korea: 1,000/5,000/10,000/50,000 won. In the registration phase, an animation was shown in which the four different banknotes were randomly hidden in the five household items. After all the banknotes were hidden, the participant was shown the location where the money was hidden twice. This test was administered after completion of the recognition test of "Speaking sentence." Each banknote was shown in random order and the participant was instructed to touch the household item where the money was hidden. They were awarded five points per correct answer. |

(*Continued*)

**Table 1.** (Continued)

| Item | Content |
|---|---|
| 7) Read and act (Language) | The sentence "Press (circle) (twice)" command was presented on the screen, and the participant was instructed to read the sentence on the screen. The words read by the participant were automatically scored using ASR. Next, a circle and triangle were displayed on the screen, and the participant was instructed to perform the sentence according to the previously read sentence. If the participant touched the circle, they were awarded five points for providing the correct answer. |
| 8) Speaking words (Execution (word fluency)) | The participants were instructed to say as many words as possible about the topic presented on the screen. First, a practice test was conducted using the theme "fruits." In the test, the participants were instructed to say the name of the animal as many times as possible in 100 s, and they were awarded three points per correct answer. The total CogMo score was obtained as follows: 0 (0–4 items), 3 (5–7 items), 6 (8–11 items), and 10 points (12 or more items). |

ASR, automatic speech recognition.

analyses, we assessed the necessary assumptions for each statistical test, including normality (using the Shapiro-Wilk test) and homogeneity of variances (using Levene's test). To determine whether there were significant differences between the two groups of variables based on cognitive function levels, we performed the Student's t-test for continuous variables and the chi-squared test for categorical variables. Additionally, Pearson's correlation test was performed to analyze the correlation between the K-MMSE-2 score and that of each CogMo test item and the total CogMo score. The correlation coefficients were interpreted as negligible ($<0.10$), weak (0.10–0.39), moderate (0.40–0.69), strong (0.70–0.89), and very strong (0.90–1.00) [30]. Moreover, to verify the test-retest reliability of CogMo, the intraclass correlation coefficient (ICC) was evaluated, and an ICC values of $<0.40$, 0.40–0.59, 0.60–0.74, and 0.75–1.00 indicated poor, fair, good, and excellent agreements [31]. Additionally, receiver operating characteristic (ROC) curve analysis was conducted to determine the cutoff value (using the Youden Index) at which the total CogMo score distinguished the CI group from the normal cognitive group. All statistical analyses were performed using SPSS (version 25.0; IBM, Armonk, NY, USA) and MedCalc (version 22.005, MedCalc Software). Statistical significance was set at $P < 0.05$.

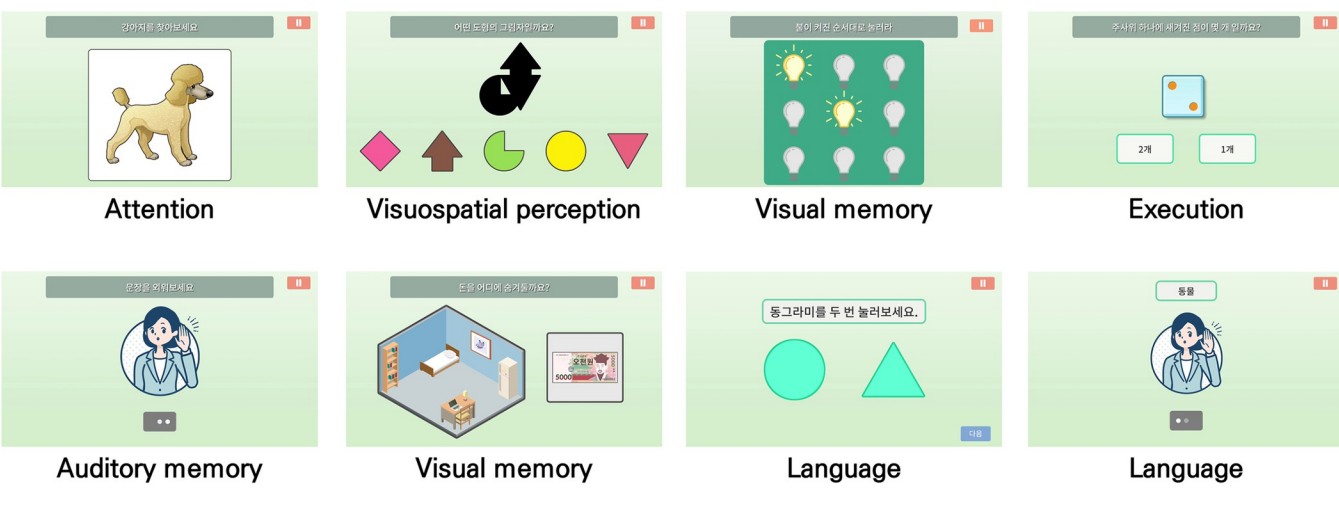

**Fig 2. Screenshots of CogMo subtests.**

## Results

### Characteristics of participants

The participants' characteristics are listed in Table 2. Their mean age was 74.6 ± 7.4 years (normal cognition group = 73.2 ± 7.3 years, CI group = 77.8 ± 6.9 years, p-value < 0.01), and no differences existed in sex (p-value = 0.32), education (p-value = 0.26), or literacy (p-value = 0.35) between the two groups according to the cognitive level. In the evaluation of cognitive function through the K-MMSE-2 score, a statistically significant difference between the normal cognition (27.4±2.8 points) and CI (20.5±3.1 points) (p < 0.001) group was observed.

### Comparison of the results of each CogMo item according to cognitive function

The results for each CogMo test item according to the cognitive function level are presented in Table 3. For all test items, including manual and automatic scoring of speech recognition, the CogMo results showed statistically significant differences between the normal and CI groups. In addition, there was a moderate-to-strong correlation between manual and automatic scoring for the following speech recognition items: speaking a sentence (r = 0.86, p < 0.001), reading and acting (r = 0.77, p < 0.001), and uttering words (r = 0.68, p < 0.001). Moreover, the total CogMo score exhibited a significant difference between the two groups for cognitive function in both manual and automatic scoring (p < 0.001), and a very high correlation between manual and automatic scoring (r = 0.98, p < 0.001).

### Correlation analysis between the MMSE and the CogMo results

The Pearson's correlation coefficients between the MMSE scores and those for each CogMo item are shown in Table 4. Among the tested items, counting numbers (Dots: r = 0.70, Dices: r = 0.77, p < 0.001) and speaking a sentence (total score, r = 0.70, p < 0.001) showed a strong correlation, and the other items showed a moderate correlation, ranging from r = 0.48 (matching shadows) to 0.65 (backward ordering of lights). The total CogMo score strongly correlated with the MMSE score (r = 0.89, p < 0.001) (Fig 3).

### Test-retest reliability of CogMo

To verify the test-retest reliability of CogMo, randomly selected participants (normal cognition group = 17, CI group = 8) were assessed again after an average of 86.2 days after the first assessment. The scores for all test items showed good or excellent agreement, except for reading and acting (ICC = 0.49, p = 0.06) and speaking words (ICC = 0.35, p = 0.15). Additionally, the total CogMo score was 62.3 ± 20.6 and 64.4 ± 19.8 for the first and second measurements,

**Table 2. Participant characteristics.**

| Variables | Total (n = 100) | Normal (n = 70) | CI (n = 30) | P-value |
|---|---|---|---|---|
| Age (years) | 74.6±7.4 | 73.2±7.3 | 77.8±6.9 | <0.01 |
| Female, n (%) | 64 (64.0%) | 47 (67.1%) | 17 (56.7%) | 0.32 |
| Education (years) | 8.5±5.1 | 8.9±5.2 | 7.6±4.9 | 0.26 |
| Literacy | 93 (93.0%) | 64 (91.4%) | 29 (96.7%) | 0.35 |
| MMSE (points) | 25.3±4.3 | 27.4±2.8 | 20.5±3.1 | <0.001 |

CI, cognitive impairment; MMSE, Mini–mental state examination.

**Table 3. Comparison of items used for computerized cognitive assessment between groups according to the cognitive function.**

| Items | Scale range (points) | Total (n = 100) | Normal (n = 70) | CI (n = 30) | p-value |
|---|---|---|---|---|---|
| Finding a puppy | 0–18 | 15.9±3.6 | 16.9±2.4 | 13.6±4.7 | <0.01 |
| Matching the shadows | 0–18 | 14.9±3.2 | 15.6±2.3 | 13.3±4.2 | <0.01 |
| Ordering the lights | | | | | |
| Forward | 0–12 | 7.6±2.9 | 8.4±2.4 | 5.9±3.2 | <0.001 |
| Backward | 0–10 | 5.8±3.1 | 6.7±2.8 | 3.7±2.9 | <0.001 |
| Counting numbers | | | | | |
| Dots | 0–15 | 12.8±3.4 | 13.8±2.7 | 10.7±3.9 | <0.001 |
| Dices | 0–15 | 11.2±5.1 | 13.0±3.9 | 7.2±5.4 | <0.001 |
| Speaking a sentence | | | | | |
| Recall | 0–15 | 5.2±5.9 | 6.7±6.1 | 1.7±3.4 | <0.001 |
| Recognition | 0–15 | 11.8±3.9 | 13.3±2.5 | 8.2±4.3 | <0.001 |
| Total score (manual) | 0–30 | 20.4±9.5 | 24.0±7.1 | 12.1±9.2 | <0.001 |
| Total score (automatic) | 0–30 | 17.2±8.7 | 20.1±7.6 | 10.2±7.0 | <0.001 |
| Finding hidden money | 0–20 | 14.5±7.3 | 17.3±5.2 | 7.8±7.2 | <0.001 |
| Read and act (manual) | 0–10 | 8.8±2.7 | 9.0±2.6 | 8.3±2.7 | <0.05 |
| Read and act (automatic) | 0–10 | 7.7±3.3 | 8.4±3.2 | 6.0±3.1 | <0.001 |
| Speaking words (manual) | 0 | 31.9±11.2 | 35.6±9.2 | 23.4±10.9 | <0.001 |
| Speaking words (automatic) | 0 | 19.0±13.4 | 23.3±12.9 | 9.0±8.4 | <0.001 |
| Total CogMo score (manual) | 0–100 | 75.1±19.8 | 83.0±14.7 | 56.6±17.6 | <0.001 |
| Total CogMo score (automatic) | 0–100 | 69.7±20.3 | 78.1±15.6 | 49.9±15.9 | <0.001 |

CI, cognitive impairment.

**Table 4. Correlations between the MMSEs and scores for each CogMo item.**

| Items | Correlation coefficient r | P-value |
|---|---|---|
| Finding a puppy | 0.61 | <0.001 |
| Matching the shadows | 0.48 | <0.01 |
| Ordering the lights | | |
| Forward | 0.61 | <0.001 |
| Backward | 0.65 | <0.001 |
| Counting numbers | | |
| Dots | 0.70 | <0.001 |
| Dices | 0.77 | <0.001 |
| Speaking a sentence | | |
| Recall | 0.56 | <0.001 |
| Recognition | 0.75 | <0.001 |
| Total score | 0.70 | <0.001 |
| Finding hidden money | 0.63 | <0.001 |
| Read and act | 0.54 | <0.001 |
| Speaking words (correct answer) | 0.61 | <0.001 |
| Total CogMo score (0–100) | 0.89 | <0.001 |

MMSE, mini–mental state examination.

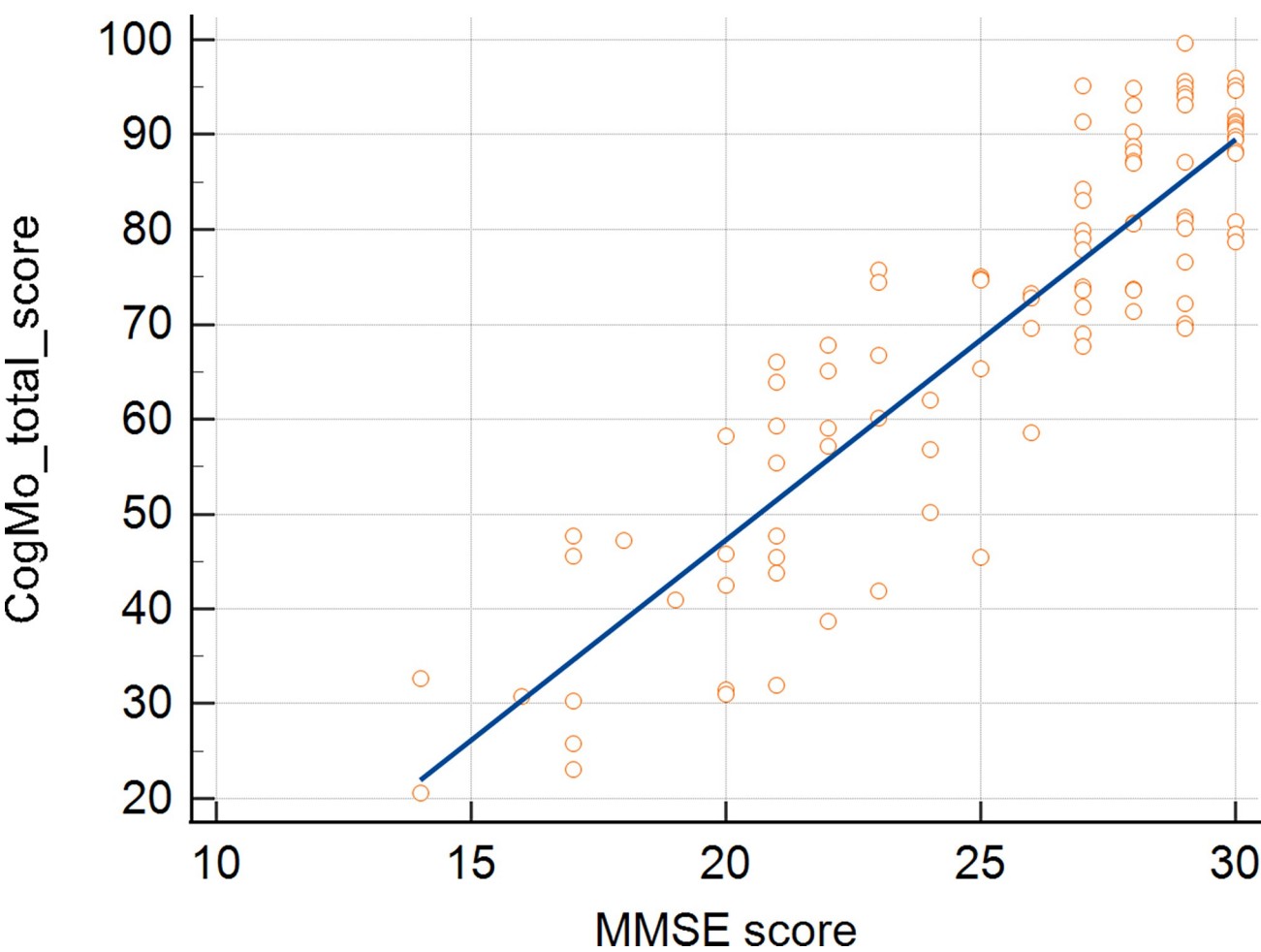

**Fig 3. Scatterplot demonstrating the correlation between the CogMo and MMSE results (r = 0.89, p < 0.001).**

respectively, and the ICC was 0.94 (p-value < 0.001), indicating a very high test-retest reliability (Table 5).

### ROC analysis for discriminating CI using CogMo

The ROC analysis for discriminating between the normal cognition and CI groups using the total CogMo score showed an area under the curve (AUC) value of 0.894 (p < 0.001), indicating high accuracy. Additionally, at the optimal cutoff value (total CogMo score = 68.8), the sensitivity and specificity were 90.0% and 82.9%, respectively (Fig 4).

### Discussion

All test items of the CogMo indicated statistically significant differences between the normal cognition and CI groups. Items using speech recognition also demonstrated significant differences between the manual and automatic scores for both groups. The MMSE and total CogMo scores were highly correlated, and the total CogMo score suggested a cut-off value to distinguish between the normal cognition and CI groups. Finally, the CogMo results confirmed that it can be used to screen and monitor CI in a community setting with a high test-retest reliability.

**Table 5. Test-retest reliability of the CogMo.**

| Items | | First measurement | Second measurement | ICC | p-value |
|---|---|---|---|---|---|
| Finding a puppy | | 14.5±4.5 | 16.4±2.1 | 0.65 | <0.01 |
| Matching the shadows | | 14.6±2.9 | 14.7±3.7 | 0.73 | <0.01 |
| Ordering the lights | | | | | |
| | Forward | 6.8±2.8 | 6.8±2.7 | 0.63 | <0.05 |
| | Backward | 5.2±2.8 | 4.9±3.1 | 0.65 | <0.01 |
| Counting numbers | | | | | |
| | Dots | 12.0±3.9 | 12.1±4.3 | 0.94 | <0.001 |
| | Dices | 10.2±5.4 | 10.1±5.2 | 0.84 | <0.001 |
| Speaking a sentence | | | | | |
| | Recall | 4.3±5.1 | 4.8±4.8 | 0.62 | <0.05 |
| | Recognition | 10.6±4.5 | 12.7±3.5 | 0.77 | <0.001 |
| | Total score | 15.1±8.6 | 17.8±7.2 | 0.80 | <0.001 |
| Finding hidden money | | 10.6±7.9 | 12.2±7.5 | 0.70 | <0.01 |
| Read and act | | 7.4±3.3 | 7.4±3.3 | 0.49 | 0.06 |
| Speaking words (correct answer) | | 14.0±13.4 | 16.2±10.6 | 0.35 | 0.15 |
| Total CogMo score (0–100) | | 62.3±20.6 | 64.4±19.8 | 0.94 | <0.001 |

ICC, intraclass correlation coefficient.

CogMo showed a high diagnostic accuracy for distinguishing the CI group from the normal cognition group (AUC = 0.894, sensitivity = 90.0%, specificity = 82.9%). In contrast, the diagnostic accuracy of the MMSE, which is widely used in clinical practice to screen for CI, varies depending on the study population and version employed [32–34]. A recent meta-analysis reported and AUC of 0.88 for the diagnostic accuracy of the MMSE for MCI screening in primary healthcare settings, which was the same as that of CogMo [35]. In addition, the MoCA, which is another cognitive function screening tool, showed a slightly lower diagnostic accuracy than CogMo (AUC = 0.846) in a meta-analysis [36]. Furthermore, the diagnostic accuracies of CCTs for discriminating CI reported thus far are similar or lower than that of CogMo, with AUC values ranging from 0.62 to 0.91 [37]. CogMo demonstrates enhanced diagnostic accuracy compared to existing CCTs by incorporating a voice recognition input interface, whereas most existing CCTs rely solely on touch-based input methods, such as touch screens or mouse/keyboard [16]. This feature allows CogMo to assess cognitive functions in a manner that closely resembles face-to-face tests commonly used in clinical practice, such as the MMSE and MoCA, while maintaining the advantages of a computerized approach. However, the automated scoring system strictly adheres to predefined rules, which may result in slightly stricter scoring compared to manual scoring. This difference can be attributed to the inherent limitations of ASR systems, such as transcription errors caused by unclear speech or dialectical variations. Despite these differences, the relative trends between the two groups based on CI remained consistent, supporting the diagnostic utility of CogMo. Additionally, CogMo can be used to evaluate various cognitive functions, such as verbal memory, language, and execution, using speech recognition. Considering these advantages, CogMo is expected to be effectively utilized for screening and continuous monitoring of cognitive functions in community-dwelling older adults.

CogMo demonstrated a high test-retest reliability, with good to excellent agreement between the test results of not only the total CogMo score (ICC = 0.94, p < 0.001) but also for most test items. Given that previous studies on the test-retest reliability of cognitive screening

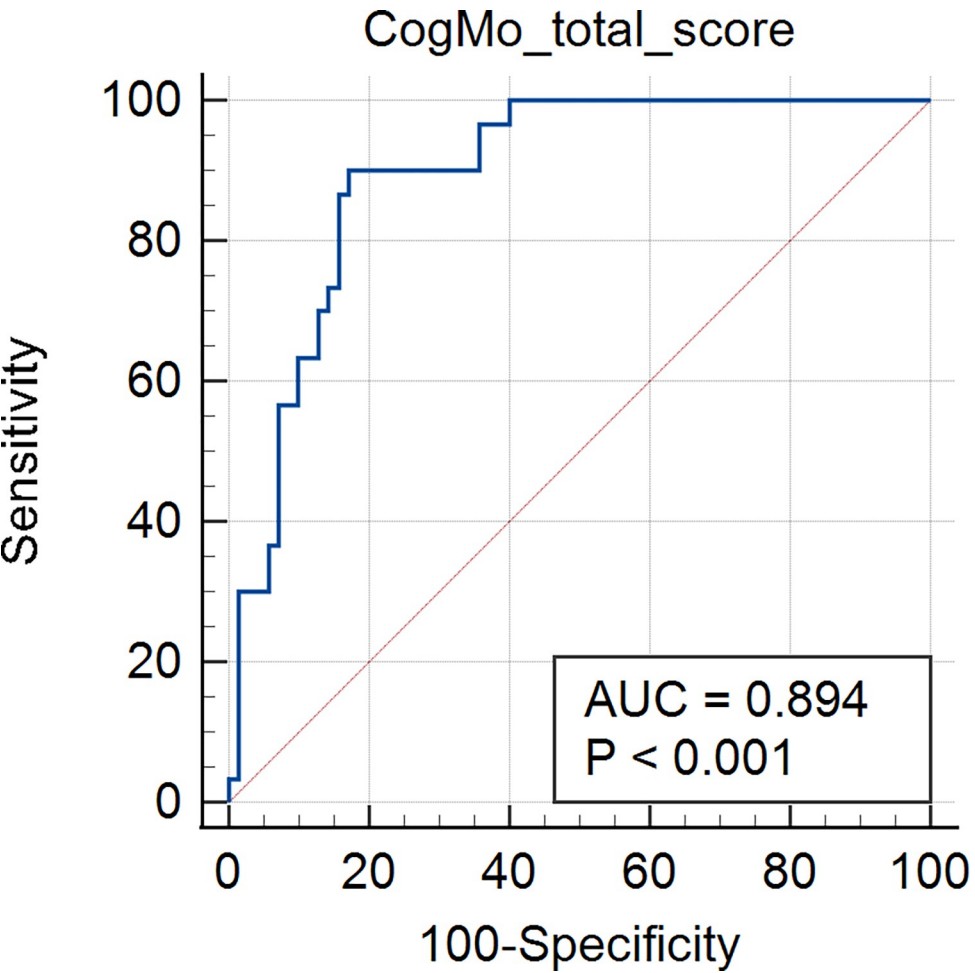

**Fig 4. Receiver operating curve analysis discriminating cognitive impairment from normal based on the total CogMo score (AUC = 0.89, p < 0.001).**

tools used in clinical practice have shown that the ICC of the MMSE ranges from 0.80 to 0.95 [38] and that the MoCA ranges from 0.87 to 0.96 [39, 40], the reliability of CogMo can be concluded to be sufficiently high for its clinical applicability. The test-retest reliability is important for cognitive function tests to ensure that changes in scores across repeated cognitive function tests reflect actual changes in cognitive function and not measurement errors due to learning differences. In particular, because CCTs such as CogMo can be self-administered to monitor cognitive functions without the need for trained technicians [41–43], a high measurement reliability over multiple tests is essential for their clinical application. Considering that the test-retest reliability (ICC) of the previously developed CCTs has ranged from 0.43 to 0.97 in previous studies [16, 43], that of the proposed CogMo was sufficiently high compared to other CCTs.

In this study, cognitive functions, such as verbal memory (speaking sentences) and execution (speaking words), were tested using speech recognition and statistically significant differences between the normal cognition and CI groups were observed. In our thorough literature review, we found few CCTs that used newly developed test items rather than clinically validated metrics for cognitive function (e.g., MoCA), which measure multi-domain cognitive functions, from visual perception to executive function, similar to CogMo [20, 43–45].

Generally, language function deficits may precede episodic memory impairment in individuals at risk of developing MCI [46]. Because existing CCTs mostly involve touching a screen interface, they can only test limited cognitive functions, such as attention and memory, using visual stimuli. However, CogMo can assess higher cognitive functions related to language, such as verbal memory and word fluency (execution), using the ASR technology. In the future, advances in digital healthcare technology are expected improve the accuracy of ASR, allowing the use of CCTs for assessing language functions in older adults.

CogMo has several advantages over existing face-to-face cognitive function assessments. First, it provides participants with pre-recorded visual and auditory instructions for performing the test, thereby minimizing errors because of instructional bias among testers [47]. Moreover, it can be administered by testers without specialized training. These advantages allow using CogMo as a tool for cognitive function screening in public health assessments and cohort studies. Second, because all data associated with the test are automatically processed and stored as personal health records, both the overall cognitive performance and changes in specific cognitive domains can be continuously monitored. This allows for the provision of personalized cognitive rehabilitation content tailored to the individual's cognitive profile based on these data. Third, as some individuals may be reluctant to have their cognitive functions assessed in a formal clinical setting, a computerized test may make them more receptive to subsequent formal clinical assessments [48]. Lastly, CogMo's ASR technology could be adapted to incorporate open-ended tasks, such as picture description, which previous studies have shown to be closely associated with CI, particularly in the language domain [49, 50]. Such adaptations could enable the evaluation of richer linguistic data and a broader range of cognitive functions, potentially enhancing its diagnostic accuracy.

This study has several limitations. First, the speech recognition accuracy of CogMo was lower than expected. However, the accuracy for all test items, except for the "Speaking Words" task, was comparable to that of other voice recognition-based CCTs [43], and the total CogMo score showed a very high correlation (r = 0.98) between manual and automatic scoring, which can be considered sufficient for clinical use. Second, because we only assessed cognitive functions using the MMSE, participants may have been misclassified between the CI and non-CI groups. We acknowledge that CI is typically diagnosed through comprehensive evaluations that combine clinical symptoms, neuropsychological tests, and other findings to reach a diagnosis confirmed by clinicians. However, this limitation was unavoidable as our study aimed to validate CogMo against the MMSE, the most widely used cognitive screening tool. We plan to conduct additional research involving comprehensive diagnostic evaluations, including neuropsychological tests, to validate CogMo's ability to detect CI as confirmed by clinicians. Finally, CogMo was developed and validated in Korean, which presents a limitation in generalizing the findings of this study to other linguistic and cultural contexts. However, certain subtests, such as finding a puppy, matching the shadows, and counting numbers, assess cognitive domains beyond language and are less dependent on the Korean language. With minor adaptations, these subtests could be applied to other linguistic groups, expanding CogMo's applicability across diverse populations.

## Conclusion

Newly developed CogMo exhibited high validity and reliability compared with the MMSE, the most commonly used neuropsychological test in clinical practice. Therefore, CogMo can be used to assess and monitor multi-domain cognitive functions in community-dwelling older adults, including those who are illiterate.

## Acknowledgments

The author has no acknowledgments to report.

## Author Contributions

**Conceptualization:** Hyun-Ho Kong, Kwangsoo Shin, Dong-Seok Yang, Aryun Kim, Hyeon-Seong Joo, Min Woo Oh, Jeonghwan Lee.

**Data curation:** Hyun-Ho Kong, Aryun Kim, Hyeon-Seong Joo.

**Formal analysis:** Hyun-Ho Kong, Aryun Kim.

**Funding acquisition:** Hyun-Ho Kong, Jeonghwan Lee.

**Investigation:** Dong-Seok Yang.

**Methodology:** Hyun-Ho Kong, Aryun Kim.

**Project administration:** Kwangsoo Shin, Dong-Seok Yang.

**Resources:** Kwangsoo Shin, Dong-Seok Yang, Aryun Kim, Hyeon-Seong Joo, Jeonghwan Lee.

**Software:** Kwangsoo Shin, Dong-Seok Yang, Hyeon-Seong Joo, Min Woo Oh.

**Supervision:** Hyun-Ho Kong, Dong-Seok Yang.

**Validation:** Dong-Seok Yang, Hyeon-Seong Joo.

**Visualization:** Hyun-Ho Kong, Min Woo Oh.

**Writing – original draft:** Hyun-Ho Kong.

**Writing – review & editing:** Min Woo Oh, Jeonghwan Lee.

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
