## [Decision Letter · Decision Letter 0]

15 Oct 2024

PONE-D-24-21463Development and validation of a self-administered computerized cognitive assessment based on automatic speech recognitionPLOS ONE

Dear Dr. Oh,

Thank you for submitting your manuscript to PLOS ONE. After careful consideration, we feel that it has merit but does not fully meet PLOS ONE’s publication criteria as it currently stands. Therefore, we invite you to submit a revised version of the manuscript that addresses the points raised during the review process.

**ACADEMIC EDITOR: The main issues raised include the need for clarity on ASR use in MCI populations and how speech parameters (e.g., speech rate, pauses) are handled. The term "higher cognitive functions" requires definition, and justifications for the CogMo scoring system need to be evidence-based. The methodology should detail how participants' vision and hearing were assessed, explain speech-to-text algorithms, and include a flowchart for ASR processing. The choice of MMSE over MOCA must be justified, and statistical assumptions (e.g., normality) and age differences between groups should be addressed. Clarification on handling missing values and the total ASR score calculation is needed. Finally, the diagnostic accuracy of CogMo, its ability to monitor multi-domain functions, and its advantages over traditional tests like MMSE require further justification. Please also ensure that you elaborate on your Data availability statement. **

Please include the following items when submitting your revised manuscript:A rebuttal letter that responds to each point raised by the academic editor and reviewer(s). You should upload this letter as a separate file labeled 'Response to Reviewers'.A marked-up copy of your manuscript that highlights changes made to the original version. You should upload this as a separate file labeled 'Revised Manuscript with Track Changes'.An unmarked version of your revised paper without tracked changes. You should upload this as a separate file labeled 'Manuscript'.

We look forward to receiving your revised manuscript.

Kind regards,

Tamlyn Julie Watermeyer

Academic Editor

PLOS ONE

**Journal Requirements:**

2. In your Methods section, please include additional information about your dataset and ensure that you have included a statement specifying whether the collection and analysis method complied with the terms and conditions for the source of the data.

This research was supported by the Medical Device Technology Development Program (grant number: 20014701, modular quantitative aging assessment and care service based on multiple sensors for aging in-home) funded by the Ministry of Trade, Industry, and Energy (MOTIE, Sejong, Republic of Korea).

Reviewers' comments:

Reviewer's Responses to Questions

**Comments to the Author**

1. Is the manuscript technically sound, and do the data support the conclusions?

Reviewer #1: Yes

Reviewer #2: Yes

2. Has the statistical analysis been performed appropriately and rigorously? 

Reviewer #1: Yes

Reviewer #2: Yes

3. Have the authors made all data underlying the findings in their manuscript fully available?

Reviewer #1: Yes

Reviewer #2: No

4. Is the manuscript presented in an intelligible fashion and written in standard English?

Reviewer #1: Yes

Reviewer #2: Yes

5. Review Comments to the Author

**Reviewer #1:** In this study, the authors aimed to develop the CogMo and to investigate the psychometric properties of this computerized test in 100 community dwelling older adults with and without cognitive impairment in South Korea.

The main techniques for collecting data and processing parameters are the ASR and manual scoring, however, the underlying concepts of the CogMo and the ASR are skeptical but susceptible of improvement. The main results are generally interesting and useful. The main contribution to filling theoretical and practical gaps is clear. Nonetheless, the unique and important contributions of the current study to relevant international circles are unclear. Accordingly, I have major concerns on justifications and clarity for research instruments, statistical techniques, and generalizability of the manuscript in its present form.

Introduction

- It is evident that the ASR can be utilised to MCI in pervious studies (e.g., Tóth et al., 2018; Vincze et al., 2022; Kantithammakorn et al., 2022). The authors should make more an effort to incorporate the ASR in the MCI population.

- p.88. The authors should clarify what do they mean by 'higher cognitive functions'? since it should be linked with the structure of the CogMo.

- It is obvious from the title that the new computerized test is based on the ASR, thus this technique should be introduced to the reader and also justify the practicality of the technique.

Method

- How did the authors check the normal vision and hearing of the participants?

- The authors should explain in detail concerning the speech-to-text algorithms. It is possible that when we talk about the ASR technique - it also involves with several parameters, that is, Speech rate, Articulation rate, Silent pause duration, Filled pause duration, etc.

- As per the CogMo items, justifications for giving one or two points for each item should be clearly indicated. Does the given rule depend on evidence-based research?

- Please provide details of the circumstances during testing the participants.

- Flowchart of the ASR processing should be incorporated to aid understanding.

- The justification for using MMSE instead of MOCA should be indicated because previous studies suggested that MOCA outperformed MMSE for detecting age-related physiological decline of cognitive functioning (e.g., Aiello et al., 2022; Manser & de Bruin, 2024)

Results

- The authors should provide results of basic statistical assumptions (normality, homogeneity of variances, skewness, kurtosis, etc.) before conducting main analyses.

- There is statistically significant for mean ages between two groups. How do the authors control this effect?

- Is there any missing value showing in the dataset, especially for the ASR scoring?

- A very strong correlation between manual and the ASR scoring in the current study. It would be better off if the authors could clarify how to calculate the total score for the ASR scoring. At the moment, it is only mentioned in the main text that the authors employed 'the Google Cloud’s speech-to-text application programming interface'.

Discussion

- How do we utilize the CogMo to monitor multi-domain cognitive functions? Since it relies on the overall score.

- How did you believe that CogMo showed a high diagnostic accuracy since the current study used MMSE as a gold standard instead of medical diagnostic from physicians or neurologists?

**Reviewer #2: **The author propose a self-administered cognitive test name CogMo using ASR and effectively demonstrated its usability to predict MMSE score. In total the CogMO has 8 components, and subjects are graded for each component, teh results indicate the score of each component is highly correlated with MMSE.

Below are some comments to consider

1) Data availability statement need to clearly mentioned.

2) It was a bit confusing that that is an automated system or needs human labelling, make it clear

3) How the scoring of different component will be done automatically in the future, through analysis of facial and spoken content or eye tracking devices.

4) making it fully automatic will lead to reduction in results, please discuss

5) MMSE prediction is also performed through speech of picture description tasks, please discuss and add references

6) Introduction is lacking discussion about the prediction of MMSE in other languages.

7) The limitation of study is Korean language, please discuss

8) what speech recognition models are used (Whisper is teh state of the art but requires resources), and how it affects the deployment of those models in low-resource dives like tablets or mobile phones. Please discuss

9) Also in introduction discussion, what advantages CogMO is brining for elders, is there any evidence to prove it is better than MMSE

6. PLOS authors have the option to publish the peer review history of their article (what does this mean?). If published, this will include your full peer review and any attached files.

Reviewer #1: **Yes: **Asst. Prof. Dr. Peera Wongupparaj

Reviewer #2: No

---

## [Author Response · Author response to Decision Letter 0]

28 Nov 2024

We deeply appreciate the time and effort you have devoted to reviewing our manuscript, titled "Development and validation of a self-administered computerized cognitive assessment based on automatic speech recognition." Your constructive feedback has been invaluable in enhancing the clarity and quality of our study. In response, we have thoroughly addressed each comment from the Academic Editor and Reviewers in our revised manuscript and the accompanying response documents. Below is a summary of the key revisions:

1. Clarifying ASR Use and Speech Parameters: We provided a detailed explanation of the Google Cloud’s Speech-to-Text API employed in CogMo’s ASR and outlined the ASR processing workflow.

2. Defining Key Terms and Justifying Scoring: Added definitions for "higher cognitive functions" and provided evidence-based justification for CogMo's scoring framework.

3. Methodological Enhancements: Included assessments of participants’ vision and hearing, detailed descriptions of the speech-to-text algorithms, and added a flowchart illustrating the ASR process.

4. MMSE vs. MoCA: Justified the use of MMSE as the reference standard due to its widespread application and adaptability.

5. Statistical Assumptions and Age Differences: Addressed normality checks, methods for handling missing values, and the potential impact of age differences between groups.

6. Diagnostic Accuracy and Advantages: Highlighted CogMo’s diagnostic accuracy, automated scoring, multi-domain assessment capabilities, and accessibility for older adults, including those with limited literacy.

7. Data Availability Statement: Revised the statement to clearly specify access procedures while adhering to ethical guidelines.

These revisions comprehensively address the Academic Editor’s and Reviewers’ comments, enhancing the manuscript’s overall clarity and quality. We have provided detailed responses to each specific comment in the accompanying response documents. Thank you again for your thoughtful feedback and guidance. We hope the revised manuscript meets your expectations, and we look forward to your further comments.

Kind Regards,

Min Woo Oh & Jeonghwan Lee

---

## [Editor Report · Decision Letter 1]

2 Dec 2024

Development and validation of a self-administered computerized cognitive assessment based on automatic speech recognition

PONE-D-24-21463R1

Dear Dr. Oh,

We’re pleased to inform you that your manuscript has been judged scientifically suitable for publication and will be formally accepted for publication once it meets all outstanding technical requirements.

Kind regards,

Tamlyn Julie Watermeyer

Academic Editor

PLOS ONE

Additional Editor Comments (optional):

Dear Dr Oh and team,

Thank you for addressing the Reviewer comments in turn. The manuscript is now approved.

Best wishes,

Tam
---

## [Editor Report · Acceptance letter]

5 Dec 2024

PONE-D-24-21463R1 

PLOS ONE

Dear Dr. Oh, 

I'm pleased to inform you that your manuscript has been deemed suitable for publication in PLOS ONE. Congratulations! Your manuscript is now being handed over to our production team.

Kind regards, 

on behalf of

Dr. Tamlyn Julie Watermeyer 

Academic Editor

PLOS ONE